# JQBENCH: A BENCHMARK FOR READING AND EDITING JSON FROM NATURAL LANGUAGE AND/OR EXAMPLES

**Gust Verbruggen**
Microsoft
Keerbergen, Belgium
gverbruggen@microsoft.com

**Chris Parnin**
Microsoft
Raleigh, USA
cparnin@microsoft.com

**Vu Le & Sumit Gulwani**
Microsoft
Redmond, USA
{levu,sumitg}@microsoft.com

## ABSTRACT

We introduce JQBENCH, a new benchmark for evaluating language models on JSON querying and transformation tasks, where the intent can be given specified using natural language and/or examples. Whereas JQBENCH is mainly aimed at using the jq tool, it can be used to evaluate other programming languages that query and/or transform JSON. Benchmarks are automatically created from two rich sources of data: Stack Overflow discussions (1496 instances with instructions and examples, called JQSTACK) and the Spider dataset for SQL generation from natural language (859 instances with instructions and JSON Schema, called JQSPIDER). We describe and analyze the automated pipeline for benchmark creation, and perform extensive baseline experiments on different models to analyze the complexity and failure modes. Using implicit feedback, the best model (Opus 4.1) scores 76% on the JQSTACK benchmarks and 81% on the JQSPIDER benchmarks. Additionally, we show (1) that access to the documentation surprisingly does not help, (2) jq lags behind Python, and (3) that automatic feedback (and therefore examples) is crucial. Besides the challenging benchmarks, we release 13K converted but filtered cases for training purposes.

## 1 INTRODUCTION

JSON has become the de facto standard for structured data exchange—powering web APIs, databases, event streams, and configuration files—and now underpins modern AI workflows, serving as a common input and output representation for large-language-model (LLM) inference and agentic workflows. A common subset of tasks involves queries and transformations of JSON representations, which can be performed with tools such as jsonpath or jq.

Consider, for example, a jq expression (left) that operates on a social media dataset (right) and selects all users with more than 100 followers and extracts the titles of their posts:

```
.users[]                                  {"users": [{"name": "A",
| select(.followers > 100)                           "followers": 42,
| {name, posts: [.posts[].title]}                    "posts": [{"title": "X"}]}]}
```

This short query filters on a numerical attribute and simultaneously traverses nested arrays while yielding a new structure. When given the simple task to "find all elements that are present in both the arrays" and three input–output examples like [[1, 2, 3, 4], [2, 4, 6, 8, 10]] → [2, 4], models struggle to generate correct expressions. GPT-5 arrives at

```
[[.[0][] as $item | select(.[1] | index($item))]]
```

which does not come close to the simple solution .[0] - (.[0] - .[1]).

This illustrates both the expressive power of jq and the challenges of generating such transformations from natural language, especially as the constraints grow more complex. Surprisingly, there is no benchmark that jointly captures natural-language prompts and executable JSON queries and transformations.

In this paper, we propose JQBENCH, a benchmark for JSON querying, filtering, and transformation from natural language and/or examples, with a specific focus on the jq query language. The flexible and potentially complex nature of JSON data, the variety of signals that can be part of the input specification—like natural language, examples and JSON schemas—and the expressive yet concise and relatively uncommon nature of jq make JQBENCH an interesting benchmark for different research directions: (1) prompting and agentic workflows using small and large language models, (2) fine-tuning of small language models, or even (3) symbolic inductive programming.

To this end, we collected Stack Overflow questions tagged with jq, as well as NL-to-SQL tasks from an improved version of the Spider dataset, and use automated pipelines to convert them into JQSTACK and JQSPIDER, respectively. From Stack Overflow, our pipeline distills realistic developer problems into machine-checkable tasks by extracting natural-language context, compiling candidate jq expressions and input–output examples, and uses an agent to generate and validate multiple test cases. From Spider, we automatically transform relational databases into JSON databases with associated JSON schemas and derive equivalent jq programs, enabling benchmarks that require reasoning over thousands of nested records. Together these sources yield a diverse corpus of 1496 (JQSTACK) and 859 (JQSPIDER) JSON querying and transformation tasks that combine authentic language, rich structure, and automatically verifiable solutions. Additionally, from the JQSTACK creation process, we release 3641 easier tasks and their solutions that can be used in fine-tuning research, both directly and as a seed for synthetic data generation.

Experiments on different models reveals that JQBENCH is sufficiently challenging: Highest baseline of 76% for Opus 4.1 on JQSTACK and 75% on JQSPIDER. Furthermore, the novelty of jq and the unique JSON-processing setting challenge weaker models, while complex JSON operations remain difficult even for stronger ones. We show interesting lessons learned from JQBENCH, including the potential of jq, the importance of implicit feedback based on examples, the "documentation trap" for capable models in agentic loops, and feasibility of JQBENCH as an interesting PBE benchmark.

In summary, we make the following contributions:

1. We present JQBENCH = JQSTACK ∪ JQSPIDER, a benchmark for generating jq expressions that query, filter and transform from natural language and/or examples that covers complex JSON operations over diverse real-world scenarios.

2. We develop an automated pipeline that extracts authentic tasks from Stack Overflow and Spider, synthesizes and executes jq programs and input examples, and verifies correctness through execution feedback.

3. We perform baseline evaluations on different large and smaller models, to understand and analyze properties of both JQBENCH and the current state of language models on jq. Among other experiments, we compare implicit feedback versus explicit feedback (tools), we compare the uncommon jq language versus the very common Python language, and we study the importance of the natural language instruction.

The dataset and evaluation scripts can be found at www.github.com/PidgeyUsedGust/jqBench. The training data is at www.huggingface.co/datasets/PidgeyUsedGust/jqBench.

## 2   JQBENCH

This section describes the two collections of problems that make up JQBENCH: JQSTACK (diverse problems from Stack Overflow with input and output examples) and JQSPIDER (problems adapted from the Spider dataset with large inputs and a schema). For a primer on jq, we recommend reviewing the manual (jq, 2025) and formal specification (Färber, 2024).

All prompts and agent tool signatures used in this section are shown in Appendix A. We use OpenAI's GPT-4.1, GPT-5 and Anthropic's Claude Opus 4.6 for data generation. Besides the final dataset, we release the generation pipeline and all intermediate artifacts, including candidates without a

successful conversion and cached model responses, to facilitate further research and development on JQBENCH.

## 2.1 JQSTACK

Each data point $(u, I, E)$ in JQSTACK consists of an instruction $u$, two or more inputs $I$ and one or more jq expressions $E$.

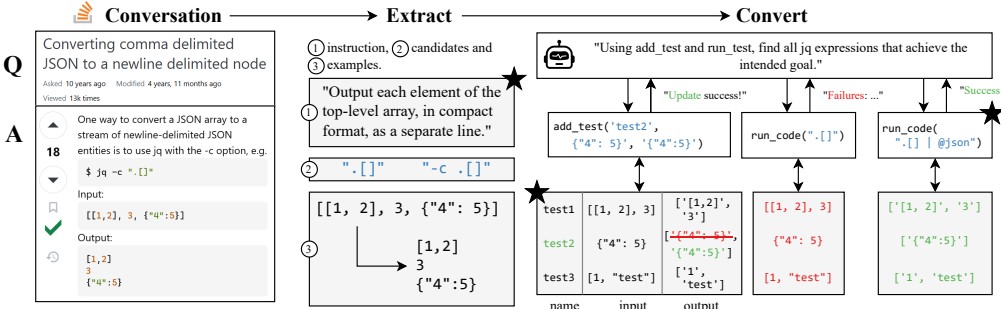

Figure 1: Overview of extraction and conversion of Stack Overflow conversations to JQSTACK tasks. Components tagged with a star (★) are part of the benchmark.

### 2.1.1 CREATION

We create JQSTACK from all Stack Overflow posts tagged with jq in four steps: (1) information extraction, (2) annotation, (3) conversion to test cases and (4) filtering. An overview of the extraction and conversion steps—the core creation process—is shown in Figure 1.

**Extraction** From each Stack Overflow discussion (question + answers) we instruct the model (GPT-5) to generate a jq task by extracting (1) direct quotes from the discussion that capture the key problem and solutions, (2) a concise and precise description of the user's intent, (3) all candidate jq expressions that satisfy this intent, and (4) optionally, any provided input and output examples. This yields 15860 raw extractions.

**Conversion** Given the intent, the candidate expressions and the candidate examples, we then instruct a model (GPT-5 into Opus 4.1) in an agentic loop to write a jq expression that satisfies the intent. The agent is given two tools: add_test(n, i, o) adds a named test n where input i is expected to produce output o and run_tests(e) runs jq expression e on all tests, providing compilation and execution results. Running add_test tool twice with the same name will update the test, which the agent can use to correct test outputs after reflecting on their results (which happened on 7.5% of conversions). All candidate jq expressions are executed on the final tests to determine successes. This yields 14620 valid conversions with GPT-5 and an additional 923 with Opus 4.6.

**Filtering** Many tasks are too trivial to challenge large and capable models, for example, a question to *print the value of a field named "text"* with a target expression .text. We therefore use a simple filter to remove tasks solvable by a jq generation prompt without interaction at temperature 0 using GPT-5 (filters 11066 cases) into Opus 4.1 (filters 1913 more cases) to keep 1496 challenging cases.

**Review** Finally, we manually review the dataset, focusing on instances that were never solved by any baseline across any evaluation. These problems were often underspecified, but conversion agent was able to brute-force them by having access to all examples (and a target solution). Common types of underspecification are hard-coded constants, adversarial edge cases, JSON schema changes (like {"a": ...} in all seen examples and {"b": {"a": ...}} in the unseen one), and ambiguity in the utterance. Ambiguity was fixed ($\sim 25$), other problems were skipped as they became too trivial with proper specification ($\sim 43$).

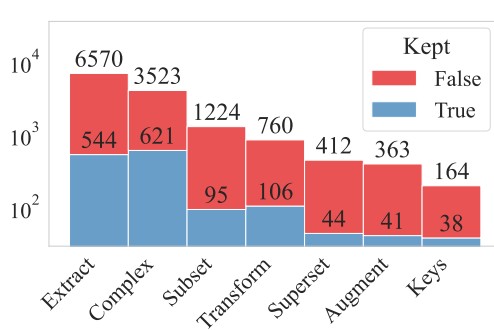

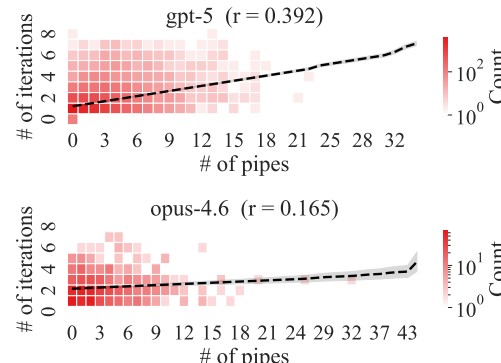

Figure 2: Distribution of failed, filtered or kept (log scale) per type of task.

Figure 3: Relation between # of iterations needed to create the task and # of | in the query.

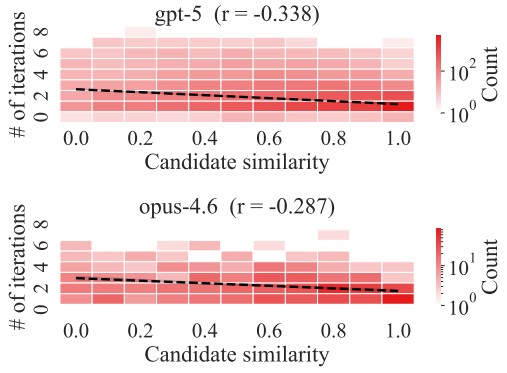

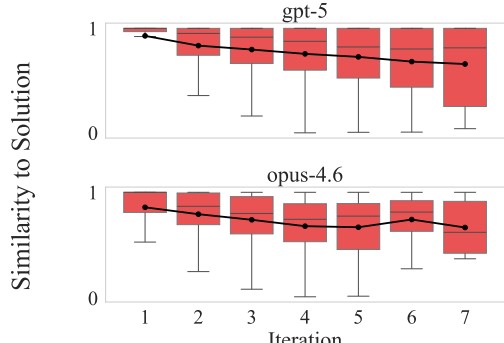

Figure 4: Similarity between candidate query and final query in function of # of iterations taken to convert.

Figure 5: With each iteration, queries suggested by the conversion argent diverge further from the original candidates.

### 2.1.2 ANALYSIS

Each task can be classified based on properties of the input and output JSON. The output can be a **subset** (removing values) or **superset** (adding values) of the input, it can be a new JSON object using only leaf values **extracted** from the input or using all leaf nodes and some **augmented** values, it can have the same structure but with some **transformed** leaf nodes, or it can be a complex transformation that fits into none of these boxes. Complex transformations typically extract nodes, transform them, and then build a new JSON object. Based on the suggested tests, Figure 2 shows the number of failed conversions, number of filtered and number of kept tasks. Most tasks involve extracting or filtering (subset) from the input JSON. Tasks that involve changing the structure (extract and complex) are notably harder, as more of them are kept.

Figure 3 shows the number of iterations needed to successfully convert the task and the number of pipe characters in the query. Other proxies for complexity (like rarity of functions, control flow, and nesting depth) show similar relations: it is harder to create more complicated benchmarks, even in the presence of candidate solutions. This result is (partially) explained by Figure 4, which shows how the number of iterations required to create a task is lower if the final solution is closer to the candidates. Finally, Figure 5 shows that suggested queries diverge further from the extracted candidates with each iteration: it does not only make small syntactic fixes, but really tries to rewrite the expression to obtain a correct solution.

## 2.2 JQSPIDER

Each data point $(u, s, e, d)$ in JQSPIDER consists of an instruction $u$, the JSON Schema $s$, a jq expression $e$, and the dataset $d$.

### 2.2.1 CREATION

We create JQSPIDER in three steps from the repaired Spider dataset (Yang et al., 2025): (1) converting each database schema to a JSON Schema, (2) converting SQL databases to JSON databases that adhere to the generated JSON schema, (3) converting SQL queries to jq expressions.

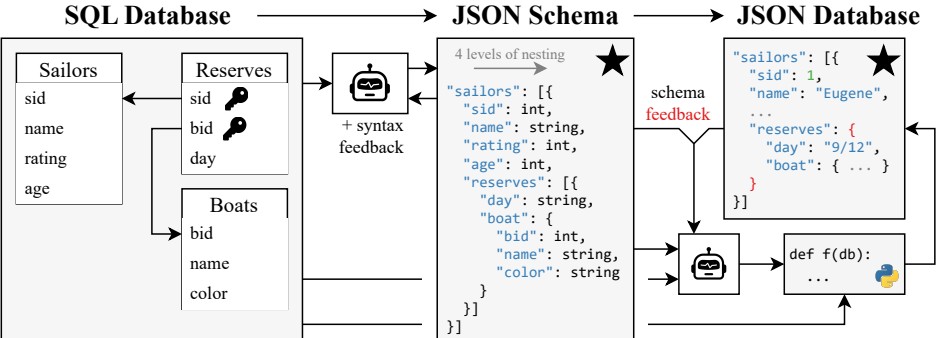

Figure 6: Overview of SQL Database to JSON Database conversion. Components tagged with a star (★) are part of the benchmark.

**JSON Schemas**   We use the model to convert a SQL schema (columns, column types and foreign keys for each table) to a JSON schema. We instruct the model to choose an appropriate *root table* and to leverage nested objects and arrays to represent one-to-one and one-to-many relations. The jsonschema Python package[1] is used to provide validation feedback. During small-scale empirical testing, we found that LLMs are able to generate more interesting schemas than symbolic heuristics.

**JSON Databases**   We then use the model to convert the SQL database to a JSON file that adheres to the generated schema. First, we symbolically generate a trivial JSON format that encodes each table separately. Second, we iteratively instruct the model to generate a Python function that converts this JSON file into a new JSON file that adheres to the given schema, using compilation and execution feedback in each iteration. Figure 6 shows an overview of this conversion. Out of 202 databases, this process succeeds for 197 of them.

**Queries**   Finally, given the natural language instruction, SQL query, JSON Schema and expected output of the SQL query on the original database, we iteratively instruct the model to write a jq expression, again using compilation and execution feedback in each iteration. This process is very similar to the conversion step in Figure 1 without the add_test tool—the tests are provided by the original SQL queries. This yields **893 benchmark tasks**.

### 2.2.2 ANALYSIS

Figure 7 shows that SQL databases with more tables yield more deeply nested JSON, generally requiring chained jq expressions. Figure 8 shows that more tables and nesting depth are both indicators of lower conversion rate. Figure 9 shows that more SQL JOINs correspond to more pipe characters in jq, confirming the need for long chains.

## 3 EXPERIMENTS

We perform extensive experiments to evaluate both our dataset(s) and the current performance of LLMs on writing jq expressions.

---
[1]https://pypi.org/project/jsonschema/

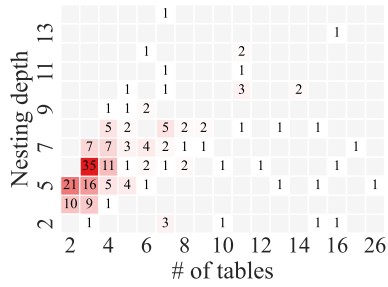

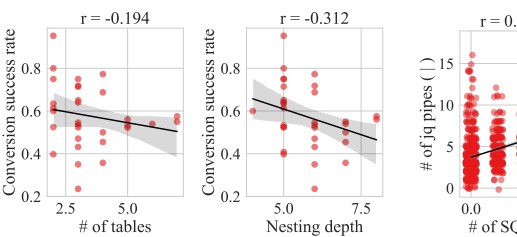

Figure 7: # of tables versus nesting depth.

Figure 8: Conversion rate versus # of tables and nesting depth.

Figure 9: JOIN versus pipes.

## 3.1 IMPLEMENTATION DETAILS AND METRICS

We use the jq Python binding[2] to execute expressions as jq.**all**(e, i). This causes all results to be wrapped in a list, for example, jq.**all**(".foo", {"foo": 1}) == [1]. We therefore provide four simple examples in the prompt (including the one above) and also consider a successful evaluation if *each* prediction {jq.**all**(e, i) == [o] (or vice versa) for the expected output o.

On JQSPIDER, the keys of record-style outputs are ignored, meaning that [{"a": 1, "b": 2}] == [{"x": 1, "y": 2}]. Unless the original query mentions ORDER BY, we also ignore order.

We use all but one of the tests as input–output examples. This strategy is common in programming-by-example (Li & Ellis, 2024).

## 3.2 BASELINES

We evaluate different models on JQBENCH in different (agentic) settings that leverage possible reward signals for feedback. The default setting (✔) is based on SELF-DEBUG (Chen et al., 2023) and simply provides feedback based on the available information. Compilation feedback is always provided, execution feedback is provided when inputs are available, and test results are provided when outputs are available. Additionally, we use the following tools in baseline experiments:

- </> run_code(e: str, i: str) runs expression e on serialized JSON object i and prints the output. This tool can be used even when inputs are not available, as the agent can synthesize its own inputs that it (thinks it) knows the output to. On JQSPIDER, for example, when only a JSON schema is provided because the whole file is too large and there is therefore no way to validate some outputs, it can synthesize smaller examples to test hypotheses.
- 🔍 search_docs(k: str[]) searches a (parsed) version of the documentation for keywords k and returns the name of all documentation sections that matches any of the keywords.
- 🖨 print_docs(s: str[]) prints the sections s of the documentation in a Markdown format. Examples of documentation sections are shown in Appendix B, and we release the parsed documentation.

In addition to the jq solvers, we compare its performance against using Python to solve JQBENCH. The model is instructed to return a Python function that accepts a single argument (the JSON object). On JQSPIDER, we instruct to return a value or a list of records to match the output format of the expected jq expression. The structure of the Python prompt mimics that of the jq prompt as close as possible (see Appendix A).

We run all setups over a maximum of eight iterations at temperature 0.

## 3.3 RESULTS

Table 1 and Table 2 show performance and configuration statistics of different configurations on JQSTACK and JQSPIDER, respectively. The following paragraphs describe these results in more detail.

---

[2]https://pypi.org/project/jq/

| MODEL | CONFIG | | | # FEEDBACK | | | # TOOLS | | | PERFORMANCE | | | |
|---|---|---|---|---|---|---|---|---|---|---|---|---|---|
| | 🄰🄱 | 🔍🖨 | ✔ | c? | e? | v? | 🔍 | 🖨 | </> | c? | e? | v? | v1 |
| opus-4.1 | Jq | | | – | – | – | – | – | 5.66 | 0.97 | 0.90 | 0.54 | – |
| gpt-5 | Jq | | | – | – | – | – | – | 0.64 | 0.84 | 0.60 | 0.15 | – |
| gpt-5-mini | Jq | | | – | – | – | – | – | 2.94 | 0.85 | 0.69 | 0.22 | – |
| gpt-5 | Jq | | ✓ | 0.48 | 0.80 | 1.31 | – | – | – | 0.98 | 0.92 | 0.68 | 0.11 |
| opus-4.1 | Jq | | ✓ | 0.09 | 0.30 | 0.67 | – | – | – | 0.99 | 0.95 | **0.76**[1] | 0.30 |
| gpt-5-mini | Jq | | ✓ | 0.69 | 1.12 | 1.60 | – | – | – | 0.96 | 0.86 | 0.59 | 0.10 |
| phi-4 | Jq | | ✓ | 5.06 | 1.26 | 0.83 | – | – | – | 0.40 | 0.24 | 0.10 | 0.02 |
| opus-4.1 | Jq | ✓ | ✓ | 0.52 | 0.11 | 0.30 | 3.60 | 2.76 | – | 0.51 | 0.46 | 0.31 | – |
| gpt-5 | Jq | ✓ | ✓ | 0.42 | 0.68 | 1.11 | 0.68 | 0.18 | – | 0.98 | 0.91 | 0.68 | – |
| gpt-5-mini | Jq | ✓ | ✓ | 0.61 | 1.03 | 1.49 | 0.74 | 0.54 | – | 0.95 | 0.84 | 0.55 | – |
| opus-4.1 | Py | | ✓ | 0.00 | 0.04 | 0.25 | – | – | – | 1.00 | 0.98 | 0.83[2] | 0.69 |
| gpt-5 | Py | | ✓ | 0.00 | 0.06 | 0.38 | – | – | – | 1.00 | 0.99 | 0.82 | 0.66 |
| gpt-5-mini | Py | | ✓ | 0.01 | 0.20 | 0.70 | – | – | – | 1.00 | 0.97 | 0.79 | 0.58 |
| phi-4 | Py | | ✓ | 0.20 | 1.13 | 1.74 | – | – | – | 0.95 | 0.82 | 0.54 | 0.34 |

🄰🄱 language, 🔍🖨 documentation mode, ✔ implicit mode, (c?) compile, (e?) execution, (v?) value, </> test expression. [1] Best overall jq performance. [2] Best overall performance.

Table 1: Results for JQSTACK under different configurations.

| MODEL | CONFIG[1] | | FEEDBACK | | TOOLS | PERFORMANCE | | |
|---|---|---|---|---|---|---|---|---|
| | 🄰🄱 | ✔ | c? | e? | </> | c? | e? | v? |
| opus-4.1 | Jq | | – | – | 1.72 | 1.00 | 0.96 | 0.77 |
| gpt-4.1 | Jq | | – | – | 0.83 | 0.99 | 0.94 | 0.75 |
| gpt-5 | Jq | | – | – | 0.00 | 0.98 | 0.91 | 0.72 |
| opus-4.1 | Jq | ✓ | 0.03 | 0.08 | – | 1.00 | 1.00 | **0.81**[1] |
| gpt-4.1 | Jq | ✓ | 0.06 | 0.12 | – | 1.00 | 1.00 | 0.79 |
| gpt-4.1-mini | Jq | ✓ | 0.01 | 0.20 | – | 1.00 | 1.00 | 0.77 |
| gpt-5 | Jq | ✓ | 0.01 | 0.11 | – | 1.00 | 1.00 | 0.78 |
| phi-4 | Jq | ✓ | 2.00 | 1.28 | – | 0.84 | 0.74 | 0.43 |
| opus-4.1 | Python | ✓ | 0.00 | – | – | 1.00 | 0.99 | 0.78 |
| gpt-4.1 | Python | ✓ | – | – | – | 1.00 | 0.99 | 0.79 |
| gpt-4.1-mini | Python | ✓ | – | – | – | 1.00 | 0.99 | 0.76 |
| gpt-5 | Python | ✓ | – | – | – | 1.00 | 0.99 | 0.80 |

🄰🄱 language, ✔ implicit mode, (c?) compile, (e?) execution, (v?) value, </> test expression. [1] Best overall performance.

Table 2: Results on JQSPIDER.

**Contrasting JQSTACK and JQSPIDER.** The two jq benchmarks reveal notably different behavior and therefore facilitate different areas of research. JQSTACK, which draws from real Stack Overflow questions, exhibits a wider spread of model performance: value-match scores range from roughly 10% (Phi 4) to 76% (Opus 4.1). In contrast, JQSPIDER—derived from Spider database queries—shows strikingly low variability: all models except for Phi 4 are in the 72%–81% range, and even Phi 4 achieves 43%. JQSTACK relies more on deep knowledge of jq, including many built-in functions (116 versus 56 different operators and functions in solutions) and the ability to define custom operators (99 versus 2 tasks where a solution defines a custom function). JQSPIDER relies less on deep knowledge of different jq operations and more on longer chains of piping map and select operations together (median of 4 pipes per task versus 3 for JQSTACK).

**Language novelty as a performance bottleneck.** The difference in performance between Python and jq on JQSTACK highlights that models often understand the task, but some do not know how to express a solution using jq. This is especially visible for smaller models (54% → 10% on Phi 4) but even GPT-5 suffers from the language bottleneck (82% → 68%). Phi 4 getting lower value feedback rates (v?) compared to GPT models is due to the fact that it does not obtain an executable expression within 8 iterations. Opus 4.1, which is known for its strong coding performance, suffers barely any

| MODEL | CONFIGURATION [1] | | | # FEEDBACK | | | # TOOLS | | | PERFORMANCE | | | |
|---|---|---|---|---|---|---|---|---|---|---|---|---|---|
| | 🅰🈁 | 🔍🖨 | ✔ | c? | e? | v? | 🔍 | 🖨 | </> | c? | e? | v? | v@1 |
| opus-4.1 | Jq | | ✓ | 0.08 | 0.31 | 0.71 | – | – | – | 0.99 | 0.95 | 0.71 | 0.33 |
| gpt-5 | Jq | | ✓ | 0.38 | 0.74 | 1.37 | – | – | – | 0.98 | 0.91 | 0.63 | 0.15 |
| gpt-5-mini | Jq | | ✓ | 0.53 | 1.10 | 1.87 | – | – | – | 0.96 | 0.85 | 0.53 | 0.14 |
| opus-4.1 | Py | | ✓ | 0.00 | 0.04 | 0.27 | – | – | – | 1.00 | 0.97 | 0.76 | 0.64 |
| gpt-5 | Py | | ✓ | 0.01 | 0.04 | 0.48 | – | – | – | 1.00 | 0.98 | 0.76 | 0.60 |
| gpt-5-mini | Py | | ✓ | 0.01 | 0.15 | 1.09 | – | – | – | 1.00 | 0.97 | 0.69 | 0.50 |

[1] 🅰🈁 language, 🔍🖨 documentation mode, ✔ implicit mode, (c?) compile, (e?) execution, (v?) value, </> test expression.

Table 3: Results for JQSTACK using only examples (no NL) under different configurations.

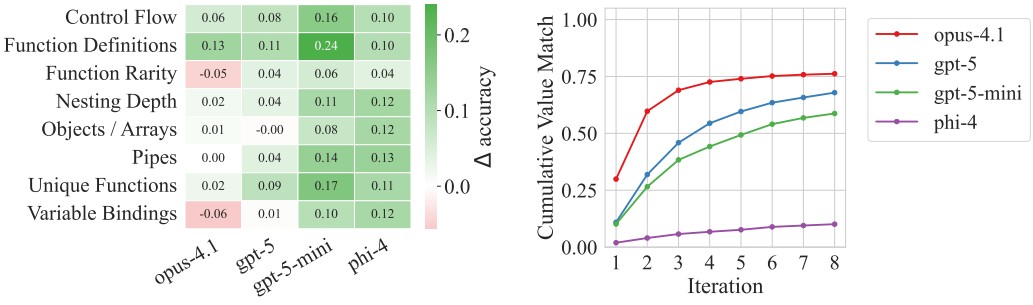

Figure 10: Heatmap of performance drops for different complexity proxies.

Figure 11: Increase of cumulative value match over iterations with implicit feedback.

performance loss (-5%). This is reinforced by its low feedback rates: only 1% of jq expressions failed to compile, and only 5% of compilable expressions failed to execute. These results highlight how JQBENCH is a useful testbed for studying how models learn unfamiliar grammars (Cassano et al., 2024; Zhang et al., 2025).

**Freedom to test is fatal.** Moving from implicit feedback to letting the model use the run_code tool sees a stark decline in performance: $-53\%$ on GPT-5 and even $-22\%$ on Opus 4.1. We identify four key reasons why letting models explicitly ask for feedback (based on the examples it sees in the prompt) is not working as expected. (1) Not using any tool calls: GPT 5 did not use any tools in 78% of tasks. (2) not (properly) leveraging the provided examples: GPT-5 (27.8%) and GPT-5 mini (28.2%) have significantly lower input coverage than Opus 4.1 (68.5%). (3) Ignoring incorrect outputs: GPT 5 (33.8%), GPT 5 mini (44.3%) and Opus 4.1 (19.9%) all use an example and get value feedback, yet propose wrong solutions. (4) Failing to generate correctly serialized JSON object for inputs or fabricates inputs: GPT-5 (20.2%). Examples are predicting `["foo":"bar"]` or repetitively predicting the target string `"1,2\n3,4"` instead of the escaped version `"\"1,2\n3,4\""`.

**Feedback is crucial.** Table 1 shows the value match after iteration 1, and Figure 11 shows the cumulative value match after each iteration. All models require a significant amount of feedback, the effect of which diminishes after 4 iterations (Opus 4.1) or 7–8 iterations (all others). Figure 12 further details the type of feedback that is given after each iteration. Opus 4.1 understand the syntax better and is therefore able to focus on the values much sooner. Stagnating performance is due to the main failure mode: overfitting on the examples (even if the utterance is explicit about generality, such as "*objects at any dept*"). The smaller Phi model struggles to generate queries that compile, even after eight iterations, indicating a promising direction for fine-tuning on the 13K filtered cases.

**What makes complexity?** Figure 10 shows the performance drop for different complexity proxies: # of {control flow, function definitions, object and array initializations, pipe characters, variable bindings, unique functions}, average function rarity and nesting depth. The performance drop is computed by diving a result across the median complexity proxy and computing the difference

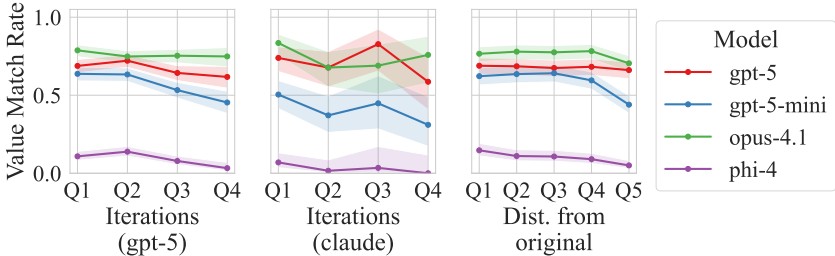

Figure 12: Feedback over iterations for different models.

Figure 13: Value match in function of number of iterations needed to create task (left and middle) and in function of distance from the original candidate solution (right).

between the value match for the "simple" and "complex" side. Opus 4.1 only affected by the # of function definitions, which are typically required in challenging and nested problems. Function rarity is not a problem.

**Contamination or complexity?**   Figure 13 shows the value match in function of the iterations taken to build each task for GPT 5 (left) and Opus 4.6 (middle), as well as the distance of the solution query to the candidate query (right). Tasks that are harder to create are harder to solve. Except for the completely changed queries—which were also harder to create and are thus likely hard tasks in general—there is no relation between the distance to the candidate solution, indicating that contamination is not an issue.

**The documentation trap.**   Opus 4.1, which the most proficient at jq according to JQBENCH, performs worse when having access to the documentation ($76\% \rightarrow 31\%$). The primary cause is getting stuck in a loop of requesting documentation, rather than solving the problem. This is can be observed by the 5x documentation request rate by Opus 4.1 compared to next most eager model (GPT 5 mini). After a certain level of proficiency, using SELF-DEBUG pays off more than doing retrieval-augmented generation over the documentation (Zhou et al., 2023).

**JSON by example.**   Omitting the natural language instruction converts each task into a programming-by-example (PBE) task, where the goal is to learn a program $p$ such that $p(\texttt{i}) = \texttt{o}$ for all example pairs $(\texttt{i}, \texttt{o})$. PBE is a popular area of research on both symbolic (Gulwani, 2011; Cropper, 2019) and neural fronts (Chen et al., 2018; Shi et al., 2022) with recent attention to LLMs (Li & Ellis, 2024). When instruction simply states that "*the query should match the following (input JSON $\rightarrow$ output JSON) examples*," the value match changes as $76\% \rightarrow 71\%$ (Opus 4.1), $68\% \rightarrow 63\%$ (GPT 5) and $59\% \rightarrow 53\%$ (GPT 5 mini). Whereas a significant decrease—the instruction is expected to provide a strong signal—these results indicate that JQSTACK poses an interesting PBE benchmark.

## 4   RELATED WORK

### 4.1   NL-TO-CODE BENCHMARKS

The closest related benchmarks to JQBENCH are DOCSPIDER (Özer et al., 2025) and JSON-SCHEMABENCH (Geng et al., 2025), which both target natural-language interfaces for document or schema-centric JSON tasks, but with different emphases. DOCSPIDER adapts the Spider text-to-SQL

dataset to document databases by converting relational data into MongoDB collections and pairing natural-language questions with MongoDB queries. Like DOCSPIDER, we also translate SQL to another representation (`jq` expressions). However, JQBENCH additionally draws on real-world Stack Overflow questions and a wide variety of organically shaped JSON inputs paired with `jq` expressions. This yields tasks that span ad-hoc filtering, restructuring, and rich transformations across highly diverse and variably nested JSON, going far beyond the relatively homogeneous datasets that underpin DOCSPIDER. JSONSCHEMABENCH evaluates constrained decoding methods for reliably generating JSON outputs that comply with complex schemas. Like JQBENCH, it can be used to assess whether language models respect schema constraints and reason about schema adherence.

Other benchmarks combine natural language with domain-specific expression languages, such as MONGOSH query expressions (MongoDB Education AI, 2025) or Vega-Lite visualization specifications (Luo et al., 2021).

Finally, several benchmarks target NL-to-CLI programs (`jq` is first and foremost a CLI tool). Terminal-Bench (Team, 2025) is a benchmark for evaluating the ability of agents to operate in terminal environments, including tasks such as *building an initramfs for a kernel*, but targets a small number of complex, multi-step system-administration tasks, with limited data manipulation tasks. NL2Bash (Lin et al., 2018) contains 12K one-line Linux shell commands—such as `top -p $(pgrep -d',' http)`—mined from Stack Overflow posts. It only contains 2 `jq` commands, however.

## 4.2 BENCHMARK CONSTRUCTION

Constructing high-quality benchmarks is challenging. For example, Yang et al. (2025) found that the widely used Spider dataset (Yu et al., 2018) contains over 30% incorrect NL-to-SQL mappings, highlighting how manual curation can introduce substantial errors. It tempting to avoid these issues by using purely synthetic data; however, Fürst et al. (2024) finds model robustness can suffer when datasets omit authentic user queries.

Researchers have drawn on several approaches for mitigating issues in benchmark construction. Benchmarks, such as ODEX, draw upon natural language queries from Stack Overflow to build a NL-python benchmark, but with manual test case construction(Wang et al., 2023). BIGCODEBENCH (Zhuo et al., 2025) used ODEX as a seed, to synthetically generate more (instruction, code) solutions in an LLM + human annotation loop. Early work used weak supervision, learning logical forms from question–answer pairs without full annotations (Berant et al., 2013). More recently, researchers have leveraged techniques that include program execution–based filtering, self-consistency checks (Wang et al., 2023), and LLM-driven self-refinement (Madaan et al., 2023).

## 5 LIMITATIONS

JQBENCH has a few limitations. We currently supports only a single JSON input per task, and do not handle multiple-input or multi-file `jq` programs, though future extensions could relax this restriction to better capture real-world multi-source workflows. The automatic nature of JQBENCH construction, which relies on an LLM to generate a solution from a rich input signal, may have limited the number of challenging tasks in the dataset. We can expand any remaining failed automatic conversions with human annotations and stronger models. Based on the English data sources, JQBENCH is available in English only.

## 6 CONCLUSION

We present JQBENCH, an automatically constructed benchmark for evaluating language models on JSON querying and transformation tasks. The benchmark includes diverse and challenging real-world problems drawn from Stack Overflow (JQSTACK) and Spider (JQSPIDER) datasets, and supports both NL and PBE tasks in a low-resource language setting. A baseline set of experiments reveal novel insights into effects of output language used for inference, potentially adverse effects of tools in agentic workflows, struggles with novel languages in small language models. Finally, our benchmark will enable future research in execution-guided decoding, exploratory data analysis, language novelty, and tool proficiency.

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

## A    PROMPT

All prompts are a minor variation on the following. The examples are specifically kept trivial: we are testing the model and not our ability to write a prompt.

```
You are a helpful assistant that helps users write jq queries.
Wrap the suggested jq query in <jq></jq> tags without surrounding quotes.

# jq Details

The jq will be executed using the Python bindings for jq with the
 `jq.all(expression, input)` function.
This causes the output to be wrapped in a JSON list, even if the result is
 a single value.
Note that you should not provide this code: you should only provide
 the jq expression itself.

Some examples:
- <jq>.foo</jq> is executed as `jq.all(".foo", {"foo": 1}) == [1]`
- <jq>{b: .a + 1}</jq> is executed as `jq.all("{b: .a + 1}", {"a": 1}) == [{"b": 2}]`
- <jq>.[0]</jq> is executed as `jq.all(".[0]", [{"a": 1}, {"b": 2}]) == [{"a": 1}]`
- <jq>map(.[0])</jq> is executed as `jq.all("map(.[0])", [[1, 2], [3, 4]]) == [[1, 3]]`
```

## B    DOCUMENTATION

### B.1    addpath.md

```
# `addpath(pathArray)`

Ensures path exists, creating objects/arrays; returns modified input.

## Example 1

**Command**: `jq 'addpath(["a",0,"b"])'`
**Input**: `{}`
**Output**: `{"a":[{"b":null}]}`
```

### B.2    optional-object-identifier-index.md

```
# `.foo?`

Optional object identifier index. Like `.foo` but suppresses errors when the input is
↪   not an object; produces `null` instead (and when iterated inside arrays, missing
↪   values simply vanish if further filtered).

## Example 1

**Command**: `jq '.foo?'`
**Input**: `{"foo": 42}`
**Output**: `42`

## Example 2

**Command**: `jq '.foo?'`
**Input**: `{"bar": 1}`
**Output**: `null`
```

```
## Example 3

**Command**: `jq '[.[] | .a?]'`
**Input**: `[{}, {"a":1}]`
**Output**: `[1]`
```

