# OpenReview forum: "jqBench: a benchmark for reading and editing JSON from natural language and/or examples"
_ICLR.cc/2026/Conference — ICLR 2026 Poster_

### Official Review · Reviewer_Vcr8 · 2025-10-27

**Soundness:** 3
**Presentation:** 3
**Contribution:** 3
**Rating:** 6
**Confidence:** 4

**Summary:**

This paper introduces JQBENCH, a new benchmark designed to evaluate the ability of large language models (LLMs) to query and transform JSON data. The tasks can be specified using natural language, input/output examples, or both. The benchmark is novel in its focus on the jq query language, a powerful but specialized tool.

**Strengths:**

1. The paper tackles a highly relevant and practical problem. As LLMs are increasingly used in agentic workflows and data processing pipelines, their ability to handle JSON—the de-facto standard for data exchange—is critical. Focusing on jq is a novel choice that tests a model's ability to learn a concise, powerful, and non-mainstream domain-specific language.
2. The creation of two separate benchmarks (JQSTACK and JQSPIDER) is a major strength. JQSTACK provides "in-the-wild" problems from real developers, ensuring the tasks are diverse and practically relevant. JQSPIDER provides structured, complex, and deeply-nested queries, effectively testing a model's logical reasoning and ability to chain operations (mapping JOINs to jq pipes).

**Weaknesses:**

1. How do you handle complex multi-table reference scenarios in a database when building jq-spider set. For example, what is the core principle for selecting the root table?
2. A key challenge with JSON objects lies in their nested structure—an issue particularly prominent in real-world use cases like agent function calls. However, the paper does not provide detailed analysis of this aspect. In particular, it fails to explore critical questions: whether the depth of nested structures impacts model performance, and if so, how.
3. The presentation of Tables 1 and 2 is confusing, with several unclear elements. For example, some accuracy values exceed 100%—a result that requires explanation, as accuracy is typically constrained to a 0%–100% range. Additionally, the entry for "gpt-4.1-mini" is split across two lines, which disrupts readability. Further questions arise from the data itself: why does GPT-5 exhibit performance numbers similar to GPT-4? And under certain configurations, why does GPT-4.1-mini outperform both GPT-5 and GPT-4.1?
4. The automated pipeline relies heavily on GPT-4.1 to generate JSON schemas from SQL and to translate SQL queries into jq expressions. This could introduce a bias where the "correct" jq solutions in JQSPIDER reflect GPT-4.1's specific style of jq programming, which may not be the most optimal or human-idiomatic solution.
5. This paper lacks a detailed error analysis of different models. For example, the paper states the cause is the model "getting stuck in a loop of requesting documentation," but doesn't deeply analyze why. Is this a failure of the agent's reasoning, a problem with the RAG prompt, or an issue with the quality/structure of the provided documentation? A few qualitative examples of these failure loops would significantly strengthen this claim.

**Questions:**

1. The paper mentions that "In contrast, JQSPIDER—derived from Spider database queries—shows strikingly low variability: all models except for Phi 4 are in the 72%–81% range." For my own suggestion, I think the Spider dataset has had potential data contamination issues since its creation. Therefore, I recommend using the BIRD [1] dataset and even its LiveSQLBench [2] variants instead. This switch might bring about some new insights.
2. In the Spider dataset, certain databases contain tables that lack "value" (i.e., have no actual data entries). How would you handle this scenario?


[1] https://bird-bench.github.io/
[2] https://livesqlbench.ai/

---

> ### Author Response · Authors · 2025-11-28
>
> > **A key challenge with JSON objects lies in their nested structure—an issue particularly prominent in real-world use cases like agent function calls. However, the paper does not provide detailed analysis of this aspect. In particular, it fails to explore critical questions: whether the depth of nested structures impacts model performance, and if so, how.**
>
> Classical measures of complexity—number of (unique) functions, average function rarity, length of the expression, number of pipes, depth of input, and proportion of expression changed with respect to original SO post—were not found to correlate with performance for the stronger models (Claude). The weaker the model, the stronger these correlations become, as shown in the below figure (correlation between complexity metrics and correct value match for the Implicit feedback mode).
>
>  https://i.imgur.com/Plro9dT.png
>
>  For strong models, the difficulty is likely not the jq generation itself (at least for larger models) but reasoning about the JSON transformation. In all cases, at least one output was correct and, in most cases, only the unseen example was wrong. Some of these are due to the specification being incomplete (which we are resolving through manual validation—see next question) but the main challenge is proper generalization to unseen cases due to the flexibility of JSON: nesting structures, data types, null handling, and answer consistency. We will extend our evaluation with these main failure modes.
>
> > **How do you handle complex multi-table reference scenarios in a database when building jqSpider set. For example, what is the core principle for selecting the root table?**
>
> The LLM chooses the root table, as it does for the whole schema. We manually validated those schemas, and it did a significantly better job than our earlier attempts at manual heuristics.
>
> > **The presentation of Tables 1 and 2 is confusing, with several unclear elements. For example, some accuracy values exceed 100%—a result that requires explanation, as accuracy is typically constrained to a 0%–100% range. Additionally, the entry for "gpt-4.1-mini" is split across two lines, which disrupts readability. Further questions arise from the data itself: why does GPT-5 exhibit performance numbers similar to GPT-4? And under certain configurations, why does GPT-4.1-mini outperform both GPT-5 and GPT-4.1?**
>
> None of the accuracy values exceed 100%. The “# feedback” and “# tools” columns are not accuracy: they are the number of times that feedback was given in the conversation and the number of tool calls. GPT-4 and GPT-5 obtaining similar performance on Python is due to these models not generalizing beyond the examples given. Both are adequate to write the code but fail to understand the meaning of the transformation. Some of these cases are false positives due to the specification being incomplete, which we are resolving with a full manual pass over the tasks. GPT-4.1-mini performs better than GPT-4 .1 within the same  configuration if that configuration heavily relies on tool calling (no implicit feedback, documentation).
>
> > **The automated pipeline relies heavily on GPT-4.1 to generate JSON schemas from SQL and to translate SQL queries into jq expressions. This could introduce a bias where the "correct" jq solutions in JQSPIDER reflect GPT-4.1's specific style of jq programming, which may not be the most optimal or human-idiomatic solution.**
>
> Since we use execution match, where we compare if two queries yield the same output, the syntax of the solution is not punished, and this bias should not be reflected in the output.
>
> > **This paper lacks a detailed error analysis of different models. For example, the paper states the cause is the model "getting stuck in a loop of requesting documentation," but doesn't deeply analyze why. Is this a failure of the agent's reasoning, a problem with the RAG prompt, or an issue with the quality/structure of the provided documentation? A few qualitative examples of these failure loops would significantly strengthen this claim.**
>
> It is not really a documentation loop—and we will rephrase this better. Because the model is eager to request documentation, even if it clearly does not need them, it has fewer iterations left to get execution feedback. We can add a visualization of trajectories and some qualitative results. GPT-4.1-mini, which is worse at jq on its own, does succeed more often with the documentation (47% to 59%) and indicates that the RAG and documentation itself is working. (Not that we did not write a RAG prompt, but the model uses the search and print tools to explore the documentation on its own.)

---

> > ### Author Response · Authors · 2025-11-28
> >
> > > **The paper mentions that "In contrast, JQSPIDER—derived from Spider database queries—shows strikingly low variability: all models except for Phi 4 are in the 72%–81% range." For my own suggestion, I think the Spider dataset has had potential data contamination issues since its creation. Therefore, I recommend using the BIRD [1] dataset and even its LiveSQLBench [2] variants instead. This switch might bring about some new insights.**
> >
> > Both BIRD and LiveSQLBench are significant improvements over Spider. Both focus on massive databases, which are (hopefully) out of the scope of what is realistic for JSON. If their queries are also more complex in terms of which operations they require, converting a realistic subset of these benchmarks to JSON would be an interesting extension for future work.
> >
> > > **In the Spider dataset, certain databases contain tables that lack "value" (i.e., have no actual data entries). How would you handle this scenario?**
> >
> > The conversion process yields an empty dataset, and any query will fail to execute, failing the conversion process. Implicitly, these are thus ignored.

---

### Official Review · Reviewer_1PiN · 2025-10-31

**Soundness:** 2
**Presentation:** 3
**Contribution:** 3
**Rating:** 4
**Confidence:** 3

**Summary:**

The paper introduces jqBench, a new benchmark for testing language models on JSON querying and transformation tasks using natural language or examples. jqBench is automatically generated using Stack Overflow and Spider dataset.

**Strengths:**

Strengths

S1. The standardization using JSON schema + jq can be very significant. And this paper propose such a benchmark in the area

S2. The pipeline automates the dataset generation with task translation+ react style verification.

**Weaknesses:**

Weaknesses

- The justification of the significance of jq operations is not enough. The comparison between the methods using table schema + SQL queries and those using JSON schema + jq is missing. Will jq outperform or is on-par with SQL generation while have the benefits of universal representation? What are the benefits of this translation.

- The tasks are mostly code generation. How about other tasks that highly rely on doc understanding, like RAG and natural language Q&A, how general the idea could be.

- The volume is not a big, less than 2000 tasks.

**Questions:**

Q1. What are the main failures observed during benchmarking?

Q2. Any human checks on the quality of the data?

Q3. How will it change if using BIRD dataset which is more complex?

---

> ### Author Response · Authors · 2025-11-28
>
> > **The justification of the significance of jq operations is not enough. The comparison between the methods using table schema + SQL queries and those using JSON schema + jq is missing. Will jq outperform or is on-par with SQL generation while have the benefits of universal representation? What are the benefits of this translation.**
>
> While both SQL and jq are used to query data, they operate on fundamentally different data models and require different reasoning capabilities. SQL queries target flat, schema-defined relational tables, whereas jq transformations operate over heterogeneous, nested JSON structures requiring structural navigation, pipeline composition, and dynamic schema inference. As a result, jq tasks cannot be reduced to SQL queries, and jq evaluation reveals different classes of LLM reasoning errors. Particularly in handling nested JSON, missing keys, and object restructuring, making jq benchmarks complementary rather than redundant with SQL-based benchmarks.
>
> > **What are the main failures observed during benchmarking?**
>
> That is a very interesting question.  For larger models (GPT-5 and Claude) the main failure mode is not the inability to write jq, but the ability to generalize the problem across different tests. In all cases, at least one output was correct and, in most cases, only the unseen example was wrong. Some of these are due to the specification being incomplete (which we are resolving through manual validation—see next question) but the main challenge is proper generalization to unseen cases due to the flexibility of JSON: nesting structures, data types, null handling, and answer consistency. We will extend our evaluation with these main failure modes.
>
> > **Any human checks on the quality of the data?**
>
> We iteratively performed human checks on data quality while improving the prompts and pipeline—even more so during the regeneration (see previous answer). Because we generate both code, input and expected output, we know that a correct solution can be obtained (given the right specification). During inspection of failure modes, we still encountered some benchmarks (and tests) that can be improved, and so we are now committing to fully validating each case before the final release. The most common issue is the instruction requiring some hyperparameters that are not inferable from all but one example, causing an incomplete specification (like *“filter values given some list”* where the list is not given and has to be inferred from data). Besides validating the current jqStack, we will manually validate and fix failed conversion cases, which will yield more complex tasks. The current version still supports our findings that (1) JSON is an interesting and complex domain for code generation from natural language and/or examples, (2) the documentation trap, (3) that weaker models suffer (more) from the language bottleneck.
>
> > **The tasks are mostly code generation. How about other tasks that highly rely on doc understanding, like RAG and natural language Q&A, how general the idea could be.**
>
> Our benchmark is a code generation benchmark. However, consider this scenario, where an agentic system is performing a task on a very large JSON document derived from an Excel Spreadsheet, containing rich data beyond values: formulas, pivot tables, charts, conditional formatting rules, slicers, data validation, etc. For that agent to understand that document, it would need to perform arbitrary queries and demonstrate dynamic schema understanding to perform downstream tasks such as manipulation and understanding. It is possible to expand the complexity of the benchmark by introducing more complex scenarios and schemas.
>
> > **How will it change if using BIRD dataset which is more complex?**
>
> Larger Text-to-SQL benchmarks (like BIRD) are largely more complex due to the sheer volume of data (549K rows per dataset versus 2K for Spider). This would make any query that requires knowledge of the data immediately more complex. (But: there are (hopefully) limits to how much data should be stored in JSON, and we are not proposing jq to replace SQL.) Extending jqBench to more benchmarks (BIRD) or more domains (mongosh) is left for future work.

---

### Official Review · Reviewer_JYjj · 2025-11-03

**Soundness:** 3
**Presentation:** 3
**Contribution:** 3
**Rating:** 6
**Confidence:** 3

**Summary:**

This paper introduces JQBench, a new benchmark for evaluating systems that translate natural-language descriptions and examples into data-transformation programs written in `jq` (a lightweight yet expressive JSON processor). Given the ubiquity of JSON as a structured data format, the release of such a benchmark is both timely and valuable for research at the intersection of program synthesis, code generation, and natural-language interfaces.

JQBench comprises two complementary datasets:

- JQStack, created by curating transformation problems from Stack Overflow posts, where realistic developer questions are distilled into machine-checkable tasks. The authors extract natural-language context, compile candidate `jq` expressions and input–output examples, and automatically validate them using an agent-based test generator.
- JQSpider, adapted from the Spider text-to-SQL dataset, reformulated here to express natural-language–to–JSON-transformation problems rather than SQL query generation.

In total, JQBench includes 751 JQStack and 893 JQSpider tasks, plus 3,641 easier subtasks derived from JQStack examples. Empirical evaluation shows the benchmark’s difficulty: even the strongest models, such as Claude 4.1, achieve only 77 % accuracy on JQStack and 81 % on JQSpider, highlighting substantial room for progress. The authors also report interesting diagnostic findings such as reduced performance when models have access to external tool feedback, and the observation that the novelty of the `jq` language itself poses a primary bottleneck.

**Strengths:**

- JQBench fills a clear gap, as despite JSON’s dominance as a structured-data representation, there are currently no benchmarks that target program synthesis for JSON transformations.

- The pipeline for JQStack, based on curated Stack Overflow problems with natural-language context and executable test cases, makes the benchmark realistic and reproducible.

- The paper documents the data-collection pipeline, validation process, and statistics, supporting reproducibility and future extensibility.

- The authors evaluate a number of models and provide thoughtful analyses of performance bottlenecks, including surprising effects such as decreased performance when models use external tools.

- The dataset’s release is likely to stimulate follow-up work on code synthesis and model grounding for low-resource programming languages.

**Weaknesses:**

- The paper’s primary contribution is the benchmark itself; there is little methodological or theoretical innovation beyond dataset construction.

- While model comparisons are informative, the analysis could be strengthened by deeper ablations, e.g., performance versus task complexity, or qualitative error analyses that categorize common reasoning failures.

- Although JQBench is diverse, the total number of problems remains modest compared to large-scale synthesis benchmarks.

**Questions:**

1. How is task difficulty distributed within JQBench? Are there systematic ways to characterize easy vs. hard transformations?
2. Given that JQBench currently supports only single-JSON inputs, have you considered extending it to include multi-step or compositional transformations that require reasoning over multiple JSON structures or files?
3. Do you plan to extend JQBench to other transformation languages or to cross-format tasks?

---

> ### Author Response · Authors · 2025-11-28
>
> > **The paper’s primary contribution is the benchmark itself; there is little methodological or theoretical innovation beyond dataset construction.**
>
> Although the benchmark itself is the primary artifact, its methodological value lies in enabling new forms of reasoning research, compositionality, structural generalization over deeply nested structures, and low-resource program synthesis, which cannot be studied using existing flat-table or SQL-based benchmarks.
>
> > **Although jqBench is diverse, the total number of problems remains modest compared to large-scale synthesis benchmarks. The volume is not big, less than 2000 tasks.**
>
> The volume of tasks is limited by real-world problems that can be automatically converted into tasks. The only way to include more tasks is to incorporate more real-world data. Based on this encouraging reviewer feedback, we have re-executed the jqStack pipeline, starting from the full Stack Overflow dump instead of samples from the API, and leveraging stronger models for conversion (GPT-5 and Claude). Starting from **15834** posts, 14620 were converted into successful cases, resulting in **12144** easy and **1235** hard tasks.
>
> > **How is task difficulty distributed within jqBench? Are there systematic ways to characterize easy vs. hard transformations?**
>
> We have performed a new complexity analysis that correlates performance with number of (unique) functions, average function rarity, length of the expression, number of pipes, depth of input, and proportion of expression changed with respect to original SO post. Based on this analysis, we find that the weaker the model, the stronger these correlations become, as shown in the below figure (correlation between complexity metrics and correct value match for the Implicit feedback mode).
>
> https://i.imgur.com/Plro9dT.png
>
> We will add this analysis to the paper, in favor of some of the existing analysis figures.
>
> > **Given that jqBench currently supports only single-JSON inputs, have you considered extending it to include multi-step or compositional transformations that require reasoning over multiple JSON structures or files?**
>
> That is a great question. Our current work introduces JSON manipulation as an interesting domain for evaluating program synthesis, and, like most benchmarks, domains and models, we plan to iteratively make our benchmark more challenging (multiple inputs, more complex tasks) and improve the strategies to solve those benchmarks.
>
> > **Do you plan to extend jqBench to other transformation languages or to cross-format tasks?**
>
> The current benchmark can be used to evaluate models’ abilities to write other transformation languages, such as jmesPath, since execution match can be performed on the final JSON. In our experiments, we use jq+Python—but any other language can be used.

---

### Author Response · Authors · 2025-11-28

We thank you for your thoughtful and constructive feedback on our submission. We are grateful for the opportunity to address your comments and clarify the contributions and ongoing improvements to jqBench.

### Strengths and impact

Thank you for acknowledging jqBench’s novelty and relevance. jqBench addresses a gap in program synthesis benchmarks by focusing on JSON transformations important to agentic workflows and data applications. Its design, grounded in Stack Overflow and Spider-derived tasks, ensures diversity and reproducibility. Our documented pipeline and analysis support extensibility and future research. We appreciate your recognition of jqBench’s potential to drive further studies in code synthesis and model grounding for low-resource languages. Based on this encouraging feedback, we have improved jqStack and our evaluation.

### Dataset expansion

In response to concerns about dataset size and diversity, and encouraged by your positive feedback, we have re-executed the jqStack pipeline. This time, we started from the full Stack Overflow dump and leveraged stronger models (GPT-5 and Claude) for the extract, conversion and filtering steps. As a result, the dataset has expanded substantially—from 1,584 posts to 14,620 successful conversions, yielding **12,144 easy** and **1,235 hard** tasks. Additionally, each task now has **at least four test cases**.

### Model update

Additionally, we have updated the jqStack experiments on this new dataset, where GPT-4.1 is replaced by GPT-5 and GPT-4.1-mini is replaced by  GPT-5-mini. (Experiments for jqSpider on new models are still running.)

https://i.imgur.com/15YkX1G.png

### New insights and analysis

* **Complexity analysis**: We now provide a detailed correlation between model performance and complexity metrics such as function rarity, expression length, pipe count, and input depth (https://i.imgur.com/Plro9dT.png). Notably, weaker models show stronger correlations with these metrics, while stronger models’ failures are primarily due to reasoning challenges rather than jq syntax.
* **Failure modes**: Our expanded evaluation identifies key issues such as incomplete specifications and generalization gaps, which we are addressing through a manual validation pass over all benchmarks.
* **Programming-by-example**: Now having at least four tests per benchmark, we have extended the programming-by-example experiments to three models (GPT-5, GPT-5-mini, Claude 4.1) and find that natural language is not as important to find a solution (roughly -2%) at the cost of requiring additional iterations (+5%). (https://i.imgur.com/H2eXgcF.png)

### Manual validation

Based on our analysis of failure modes, we have discovered a limitation of the generation process, where some tasks are underspecified (missing constants) that are overfit by the conversion process. To ensure high quality, we are conducting a full manual pass over the dataset to fix these issues. This process involves validating each case for correctness and completeness, fixing failed conversions and incomplete specifications, and adding more complex tasks where feasible. This commitment will ensure that jqBench remains a reliable and valuable resource for the research community. Note that the current version still supports our findings that (1) JSON is an interesting and complex domain for code generation from natural language and/or examples, (2) the documentation trap, (3) that weaker models suffer (more) from the language bottleneck.

---

### Meta-Review · Area_Chair_9v5S · 2026-01-08

**Summary:**

This manuscript proposes jqBench, a benchmark to evaluate LLM systems in JSON querying and transformation. jq is a domain-specific language similar to SQL but transforms JSON data entities. The benchmark systematically evaluates LLM capability (given tools and compilation feedback) in synthesizing jq programs given natural language descriptions and input-output examples.

Reviewers are, in general, positive about the work and appreciate its novel contribution as the first benchmark targeting JSON transformation program synthesis. Further, reviewers recognized the well-documented and automated & transparent dataset curation process along with the thoughtful analysis and findings. Here are the major reviewers' concerns:

1. (shared concern) Deeper ablations to strengthen analysis and more illustrations on model errors are needed.
2. (shared concern) The benchmark data size is modest.
3. (shared concern) Incorporate more complex data schema or foundation dataset beyond Spider.
4. Limited innovation beyond dataset construction.
5. jq is not significant or popular enough.
6. Heavy use of GPT4.1 in benchmark curation may incur bias.

**Reviewer Concerns:**

The reviewers provide a detailed rebuttal. However, no reviewer replied. From what I can tell:
For concern 1: The newly added analysis and rebuttal text clarifications help to mitigate the concern.
For concern 2: The benchmark data size is comparable to other existing benchmarks, so it is not a strong concern.
For concern 3: The authors admit them as a future work.
For concern 6: Ground truth as a criterion debiases the usage of GPT4.1.
Other concerns are more subjective. For example, "jq is not significant" could be viewed as a drawback due to limited practical value; but may also serve as an ideal testbed for model capabilities on learning novel programming tasks and low-resource programming languages.

Overall, I think most objective concerns are addressed; while subjective concerns may be outstanding, but it is fine.

**Reviewer Scores:**

I would anticipate that Reviewer 1PiN may increase their score from 4 to 6, since their concerns are well addressed during the rebuttal phase, except for those unique. Other review scores may not change. As a result, if reviewers had been able to participate, the scores may have converged towards the acceptance side.

---

### Decision · Program_Chairs · 2026-01-26

Accept (Poster)